# Fabrication of Polysulfone-Surface Functionalized Mesoporous Silica Nanocomposite Membranes for Removal of Heavy Metal Ions from Wastewater

**DOI:** 10.3390/membranes11120935

**Published:** 2021-11-26

**Authors:** Abdullah A. Alotaibi, Arun Kumar Shukla, Mohamed Habib Mrad, Abdullah M. Alswieleh, Khalid M. Alotaibi

**Affiliations:** 1Department of Chemistry, College of Sciences and Humanities, Shaqra University, Ad-Dawadmi 11911, Saudi Arabia; aaalotaibi@su.edu.sa (A.A.A.); m.mrad@su.edu.sa (M.H.M.); 2King Abdullah Institute for Nanotechnology, King Saud University, Riyadh 11451, Saudi Arabia; ashukla@ksu.edu.sa; 3Department of Chemistry, College of Science, King Saud University, Riyadh 11451, Saudi Arabia

**Keywords:** nanocomposite membrane, functionalized mesoporous silica nanoparticles, polysulfone, heavy metal removal

## Abstract

Membranes are an efficient way to treat emulsified heavy metal-based wastewater, but they generally come with a trade-off between permeability and selectivity. In this research, the amine and sulphonic groups on the inner and outer surface of mesoporous silica nanoparticles (MSNs) were first modified by a chemical approach. Then, MSNs with amine and sulphonic groups were utilized as new inorganic nanofiller to fabricate mixed matrix polysulfone (PSU) nanocomposite membranes using the classical phase inversion approach. The resultant nanoparticles and membranes were characterized by their physico-chemical characteristics as well as determination of pure water permeability along with cadmium and zinc ion removal. Embedding nanoparticles resulted in a significant rise in the water permeability as a result of changes in the surface properties and porosity of the membrane. Furthermore, the efficiency of developed membranes to remove cadmium and zinc was significantly improved by more than 90% due to the presence of functional groups on nanoparticles. The functionalized-MSNs/PSU nanocomposite membrane has the potential to be an effective industrial effluent removal membrane.

## 1. Introduction

Freshwater scarcity is expected to be a major global challenge in the coming decades, particularly in arid regions. As the world’s population grows, so does the demand for food and energy, necessitating the need for more water resources for drinking, household, industrial, and agricultural purposes. Due to a lack of water, extensive research has been conducted to reuse wastewater and eliminate hazardous metals before discharge, as 80 percent of wastewater is discharged without being treated or recycled [1,2]. According to the World Health Organization, billions of people are without access to clean drinking water, and every year, many children die from diseases caused by contaminated water [3]. Everyone has the right to enough, adequate, safe, and inexpensive water for residential and personal use, according to the United Nations General Assembly [4,5].

Metal-contaminated wastewater released by industry is the most common cause of pollution in water sources; metal contaminants are persistent in aquatic ecosystems and do not decompose [6,7]. Heavy metal ions, including arsenic, chromium, lead, cadmium, and zinc are among the most common contaminants found in water systems. Even at trace quantities, these heavy metal ions are harmful to humans, causing neurological difficulties, cancer, and kidney and liver damage. As a result, water contaminated with heavy metals must be purified before being released into the environment [7,8,9,10,11,12,13,14].

Traditional methods for removing heavy metal ions from effluent include coagulation, electrodeposition, flocculation, extraction, crystallization, ion exchange, adsorption, and chemical precipitation. The majority of these processes have significant drawbacks, including operating in a series of heterogeneous reactions or distributing substances between phases, which generally necessitates a long operating period and a large amount of reagent, or the release of toxic sludge that must be disposed of. Researchers are increasingly interested in heavy metal removal utilizing membrane technology because this is a low-energy, ecologically beneficial technique. The most essential aim is to design a membrane that will increase separation efficiency, especially for removing heavy metal from aqueous systems [15,16,17,18,19,20,21,22]. Nanofiltration is a pressure-driven membrane method that has just recently been created. Nanofiltration is a low-energy treatment method that has a greater permeation and better selectivity for multivalent ions than reverse osmosis. Therefore, a variety of nanofiltration membranes have been created for various separation purposes. Currently, several nanomaterials are used to develop next-generation nanocomposite membranes, and research is being conducted to improve the membrane characteristics. Nanocomposite membranes, which introduce the beneficial properties of a polymer matrix and nanoparticles, are suggested to have superior characteristics as well as provide additional strategies for preventing surface biofouling and energy demand, associated with long-term reliability and performance for freshwater stream production [23,24,25,26,27,28]. Numerous types of nanoparticles, including carbon nanotubes, and graphene oxide, aluminium oxide, zinc oxide, mesoporous silica nanoparticles (MSNs), titanium dioxide, and zirconium dioxide, have been incorporated as additives with various polymer materials, including cellulose acetate, polyvinylidene fluoride, polyetherimide, and sulfone polymers, to develop nanocomposite membranes [29,30,31,32,33,34,35,36].

MSNs have lately drawn the interest of membranologists because they have been widely accessible, cost-effective, biocompatible, are surface changeable using multiple functional groups, have a superior surface area, and are regarded to be safe materials for usage in people and animals [37,38,39]. However, van Der waals interactions between metal or oxide nanoparticles and the polymer generate inhomogeneous dispersion and aggregation, which are two of the most difficult problems to solve when producing nanocomposite membranes [40,41,42,43]. To address these issues, researchers have designed a hybrid material combining modifier chemicals and MSNs with improved properties. The modifier chemicals enhanced MSNs as well as allowed for more exposure of the surface and its functional groups to the surrounding area, resulting in improved dispersion in the polymer solution and reduced MSNs aggregation. As a result, using modified MSNs in polymeric nanocomposite membranes is predicted to result in membranes with increased hydrophilicity, improved surface characteristics, and better rejection performance [44,45,46].

In this research, we developed a nanocomposite membrane typically composed of functionalized-MSNs and polysulfone (PSU) with the aim of investigating its heavy metal removal performance. The functionalized-MSNs utilized in the study was synthesized by wet chemistry methods. The influence of the nanoadditive, functionalized MSNs, on membrane physico-chemical characteristics, porosity, and performance, i.e., water flux and rejection, was studied. The specific surface area of the functionalized MSNs nanomaterials were determined using Brunauer–Emmett–Teller (BET) surface area measurements, as well as transmission electron microscopy (TEM). X-ray photoelectron spectroscopy (XPS), scanning electron microscopy (SEM), and water content and porosity measurements were used to characterize the prepared functionalized-MSNs/PSU nanocomposite membranes. We discovered that the inner and outer surface amine and sulphonic groups functionalized-MSNs/PSU nanocomposite membrane removed more than 90% of cadmium (Cd) and zinc (Zn), while having a greater permeability than the pure PSU membrane. Notably, the effectiveness of heavy metals’ removal is influenced by surface characteristics and structure, which are influenced by the amine and sulphonic groups of MSNs. As a result, functionalized-MSNs may be utilized as a recommended nanoadditive to enhance the surface characteristics of the PSU membrane for heavy metal ion removal from industrial effluent.

## 2. Materials and Methods

### 2.1. Materials and Reagents

PSU was supplied by Solvay Advanced Polymer, Brussels, Belgium. N-Cetyltrimethylammonium bromide (CTAB, 98%), tetraethylorthosilicate (TEOS, 98%), 3-aminopropyltriethoxysilane (APTES, >98%), (3-mercaptopropyl) trimethoxy, n-hexane (HPLC grade), ammonium hydroxide (28%), cadmium nitrate (99%), zinc nitrate (99%) were provided by Sigma–Aldrich, Missouri, USA. Fisher Scientific (Massachusetts, USA) provided the hydrochloric acid (36%) and sodium hydroxide (98%). WinLab (New Jersey, USA) provided sodium azide (98%) and ammonium nitrate (99%). BDH (Dubai, united Arab emirates) provided the N,N-dimethylformamide (DMF, 98.5%) and toluene (99.5%). All of the chemicals were used exactly as they were supplied to us. Throughout the experiment, deionized water (DI) was used.

### 2.2. Surface Functionalization of Mesoporous Silica Nanoparticles

The inner and outer surface amine and sulphonic groups of MSNs were chemically synthesized in this work and presented in Figure 1. For the inner surface, a 170 mL aqueous solution of CTAB (1.0 g) was mixed with 7.0 cm^3^ of NH_4_OH and heated at 40 °C. Within 20 min, TEOS (5.0 cm^3^), APTES (0.2 cm^3^), and n-hexane (20.0 cm^3^) were added to the aqueous solution. At 40 °C, the reaction mixture was stirred for 17 h. Centrifugation was used to extract the product (N-MSNs), which was then washed six times with DI water and methanol. In the case of sulphonic groups on the outer surface, MSNs-NH_2_ was suspended for 20 h in a 100.0 cm^3^ solution of MPTMS in toluene (0.01 M). The material was centrifuged and then washed with toluene and ethanol. The material was suspended in a solution of acetic acid and hydrogen peroxide (1:1) to remove the surfactant and convert thiols to sulphonic groups. The solution was then stirred and heated at 120 °C overnight. The final product functionalized-MSNs was centrifuge separated and washed multiple times in methanol.

### 2.3. Membrane Preparation

The specific composition of casting solutions employed in this work can be seen in the preparation of functionalized-MSNs/PSU nanocomposite membranes. To completely disperse the nanoparticles into the solvent, a 1.0 wt% quantity of functionalized-MSNs was sonicated in DMF for 15 min using a digital sonifier (Branson Ultrasonics Corporation, Brookfield, Connecticut, USA). Following the dispersion of the functionalized-MSNs in the solvent, 20 wt% of dry PSU polymer pellets were dissolved in the doped solution by continuous mixing and heating at 50 RPM at 70 °C until the pellets were entirely dispersed and the solution was homogeneous. Prior to this endeavor, casting solutions were degassed for 30 min and kept under vacuum for 24 h to ensure that all air bubbles had been removed from the solution. The solution was placed onto a clean glass plate and spread out using a casting knife with a space of 100 ± 2 μm between the knife and the glass plate. To complete the phase-inversion processes, the casted film was placed in a water bath for one day at room temperature. To avoid microbiological contamination, the membranes were rinsed with distilled water and stored in a 0.2% sodium azide solution until the experiments.

### 2.4. Nanoparticles and Membrane Characterization

High-resolution transmission electron microscopy (TEM) was used to examine the morphology of functionalized-MSNs (JEOL, JEM-2100F, Tokyo, Japan). A SurPASS electrokinetic analyzer (Anton Paar, GmbH, Austria) was used to determine the charging properties of the functionalized mesoporous silica nanoparticles. Furthermore, images of the surface and cross-sectional were analyzed using scanning electron microscopy (SEM, JEOL, Japan) to investigate the morphology of the membrane samples. Dry and dust-free membrane samples were cut into various shapes and sizes and submerged in liquid nitrogen for breakdown for the SEM examination. They were then attached to sample holders (stubs) with conductive double-sided carbon—coated tape. After that, the samples were placed in a vacuum chamber for platinum sputtering, which made them electrically conducive. The surface and cross-sectional morphology of the membrane samples were studied using a 5 kV acceleration voltage. In order to investigate the statistical diameter distribution profile of nanoparticles, ImageJ processing software was utilized.

The thermal and/or oxidative stability of the membrane were determined by thermogravimetric analysis (TGA). A thermogravimetric analyzer was used to determine its compositional characteristics (Mettler Toledo, Austria). The 10 mg of membrane sample was put in a ceramic pan and examined under an inert nitrogen environment (40 mL/min) at a heating temperature of 20 °C/min under dynamic condition between 100 °C and 700 °C.

The mechanical characteristics of the membranes were investigated utilizing the LR5K Plus tensile test equipment (Llayd Instruments Ltd., Bognor Regis, UK). The tensile test has been used to see how membranes react when they are under tension. A dumbbell-shaped samples has generally pushed to its breaking point in a basic tensile test to evaluate the membrane’s ultimate tensile strength. The tensile test was performed at room temperature with a relative air humidity of 18% for this test. The average of at least three membrane samples was used to calculate the tensile strength, Young’s modulus, breaking stress, and breaking elongation for each sample. NEXYGEN Plus software was used to obtain tensile data.

A gravimetric technique was used to determine the equilibrium water content of the membranes. The membrane was 9 cm^2^ in size and was submerged in water for 24 h. After wiping off the excessive water from the surface, the wet membranes were measured. Thereafter, the wet membranes were dried in an oven at 50 °C for 24 h, and their dry weight was noted as well. Equation (1) was used to determine the water content percentage absorbed by the membranes [47].
(1)$$\mathrm{EWC}\;\left(\%\right)\;=\;\frac{\left(\mathrm{WW}-\mathrm{WD}\right)}{\mathrm{WW}}\times 100$$
where WW denotes the membrane sample’s wet weight (g), WD denotes the membrane sample’s dry weight (g), and EWC is the equilibrium water content (%).

The gravimetric technique was used to determine the total porosity of the membrane sample, as described in Equation (2) [48].
(2)$$\mathrm{Porosity}\;\left(\%\right)\;=\;\frac{\mathrm{WW}-\mathrm{WD}}{\rho \;\cdot \;A\cdot {\;l}_{m}}\times 100$$
where WW denotes the wet membrane weight, WD denotes the dry membrane weight, A denotes the membrane effective area (m^2^), 0.998 g/cm^3^ is the water density, and $l_{m}$ denotes the membrane thickness (m).

In addition, the Guerout–Elford–Ferry Equation (3) was used to compute the mean pore radius of the membranes employing porosity and water flux measurements.
(3)$$\mathrm{Mean}\;\mathrm{pore}\;\mathrm{radius}\;\left(\mathrm{nm}\right)\;=\;\sqrt{\frac{\left(2.9-1.75\varepsilon \right)8\eta l_{m}Q}{\varepsilon \cdot A\cdot \Delta P}\;}$$
where, η: water viscosity (8.9 × 10^−4^ Pa s), Q: volume of the permeated pure water per unit time (m^3^s^−1^) and ΔP: operation pressure (2 bar).

### 2.5. Membrane Performance

High-membrane performance was evaluated in pure water permeability using a pressure-driven stirred cell filtration system (Amicon–Millipore, USA), which was pressured by nitrogen gas at room temperature and had a 32 cm^2^ active membrane area. Figure 2 shows a diagram of the dead-end stirred-cell filtering system. Pure water was allowed to flow through the membranes utilizing an above dead-end filtration system to evaluate water permeation performance. To eliminate any residual solvent or unreacted polymer, membranes were compacted for 2 h at 4-bar transmembrane pressure (TMP). Water flux was studied in a steady-state condition at ambient temperature after compaction at varied transmembrane pressures ranging from 1 to 4 bar using Equation (4).
(4)$$\mathrm{Jv}=\frac{V}{A\cdot \Delta T}$$
where Jv represents the water flux (LMH), V represents the permeating water volumetric flow rate (LPH), A represents the effective membrane surface area (m^2^), and ∆T represents the sampling period (h). Water permeability was estimated using Equation (5) from the slope of the linear connection between water flux and transmembrane pressure [49].
(5)$$\mathrm{Wp}=\frac{\mathrm{Jv}}{\Delta P}$$
where ΔP is the transmembrane pressure driving force (bar) and Wp is the water permeability (LMH·bar^−1^).

Single metal ion solutions containing with concentrations of 5, 25, and 50 ppm, as well as a mixed metal ion solution with a total ion concentration of 10 ppm (5 ppm of each ion), were used to investigate the Cd^2+^ and Zn^2+^ removal performance. TMPs of 4 bar, natural pH feed solution, and ambient temperature were used during the studies. An atomic absorption spectrophotometer (200 Series, Agilent Technologies, USA) was used to detect metal ion concentrations in the feed and permeate, and a pH meter was used to assess pH values (Model 250, Denver Instrument, USA). Three rejection and solute flux tests were conducted for each membrane, with the average values presented. The percentage rejection was calculated using the following Equation (6).
(6)$$\mathrm{Rejection}\;\left(\%\right)\;=\;\left(1-\frac{\mathrm{Cp}}{\mathrm{Cf}}\right)\;\times \;100$$
where Cp denoted permeates concentration and Cf denoted feed concentration.

## 3. Results and Discussion

### 3.1. Analyze the Morphology

Transmission electron microscope (TEM) and scanning electron microscope (SEM) were used to study the surface morphology and structural characteristics of nanoparticles. The material following amination is shown in TEM images in Figure 3a. The images of the sample show sized nanospheres with center-radial mesoporous and are more uniform with separation from one another, suggesting that the MSNs morphology is unaffected by the surface amination process. The SEM result for functionalized nanoparticles, shown in Figure 3b, indicated variations of particles size, discrete spherical shape with a particle size of around 190 nm.

The pore structure of functionalized-MSNs was investigated using N_2_ adsorption–desorption isotherms, as shown in Figure 4. The specific surface area of functionalized-MSNs was calculated using the Brunauer–Emmett–Teller (BET) technique, and it was found to be 920.9 m^2^/g. The pore volume was 1.325419 cm^3^/g, whereas the micropore volume was 0.013007 cm^3^/g, according to the results. Following IUPAC classifications, the functionalized-MSNs exhibited type I and IV isotherm curves, confirming the characteristic microporous and mesoporous structure [50,51]. The existence of micropores was indicated by an almost vertical increase in the low-pressure range (P/P_0_ ~ 0). At a P/P_0_ of 0.35–0.70, the typical hysteresis loop confirms the existence of mesopores. The adsorption isotherms for P/P0 around 1.0 increase in relation to the macropores. The Barrett–Joyner–Halenda (BJH) technique was used to calculate the pore size distribution of functionalized-MSNs, which indicated that the pore size distribution was typically centered between 20 and 90 Å, with an average pore size of 57 Å. As a result, surface functionalized with amine and sulphonic groups may be readily impregnated into mesoporous silica nanoparticles when they are in close contact.

At various pH levels, the surface charge characteristics of functionalized mesoporous silica nanoparticles were investigated. The negative charge for nanoparticles surfaces at pH values ranged from 4 to 9, as shown in Figure 5. The defining properties of functional groups are primarily responsible for the substantially negative charge on the nanoparticles surface. The resultant functionalized mesoporous silica nanoparticle has a considerable influence on PSU membrane performance.

Figure 6 shows membrane morphologies in terms of cross-section and surface topographies. The cross-section morphology of the SEM of the pure PSU and functionalized-MSNs/PSU nanocomposite membranes with 1.0 wt% functionalized-MSNs content was exhibited at different magnifications in Figure 6a,b,d,e. A pure PSU membrane has asymmetrical structures, such as thin, free top layers (skin and nodular layers), and porous sublayers (such as the finger-like pores, macrovoids). In comparison to pure PSU membranes, Figure 6d,e demonstrates the addition of functionalized-MSNs to the PSU matrix, resulting in a typical composite structure with fully formed macrovoids. As shown in Figure 6e, the cross-section structures of the (1.0 wt%) functionalized-MSNs/PSU nanocomposite membranes had a high-porous structure and a dense active skin layer with nanopores and completely formed microvoids, resulting in a higher-permeability membrane. The hydrophilic characteristics of the functionalized-MSNs may be responsible for the considerably improved cross-section structure of nanocomposite membranes. Furthermore, the incorporation of nanoparticles caused the casting solution to become thermodynamically unstable, accelerating the demixing process [52,53]. The permeability values are in good accord with the SEM findings. SEM was used to examine the surface morphologies of the PSU substrate and nanocomposite membranes. The membranes’ surfaces were defect-free and smooth, as illustrated in Figure 5c and Figure 6f. EDS elemental mapping was used to further characterize the dispersion of the functionalized-MSNs in the PSU membrane. The EDS mapping of the Si element distribution is shown in Figure 7. Si appeared to be evenly distributed throughout the blended PSU membrane matrix, with no signs of aggregation or segregation on the surface. The nanocomposite membrane surface was nanoparticles aggregation free as comparison to the PSU membrane surface. When compared to the PSU membrane, the functionalized-MSNs/PSU nanocomposite membrane had better functionalized-MSNs nanoparticles dispersion in the PSU matrix and as well as excellent adhesion between nanoparticles and polymer. At a constant functionalized-MSNs concentration, the nanocomposite membrane had the greatest rejection.

### 3.2. Spectral Analysis of Membranes

The chemical compositions of the pure PSU membrane, and functionalized-MSNs/PSU nanocomposite membrane were characterized by XPS, as shown in Figure 8. The primary elemental peaks on the membrane surface were identified, dependent on the intensity of the carbon (C 1s) and oxygen (O 1s) at 285, and 534 eV, respectively, in the XPS spectra (Figure 8a,b). The spectra of the nanocomposite membrane also show the existence of one new silica (Si 2s) peak at 169 eV, which is attributed to the presence of functionalized-MSNs on the surface. Moreover, Figure 8c,d shows high-resolution XPS spectra of typical C1s core level membranes from pure PSU and nanocomposite membranes, with the peaks fitted using a linear combination of a Gaussian function and a Lorentzian function. For the blended functionalized-MSNs/PSU matrix, the C1s spectrum contains unique carbon peaks. The binding energies at ~284.2, ~285, ~285.5, and ~286.8 eV are assigned to C-Si, C-C/C-H, C-N, and C=N/C-O, respectively, which should be from the amine and sulphonic groups molecules [51]. The XPS spectra clearly show that functionalized-MSNs attach to the membrane surfaces effectively.

### 3.3. Thermal Stability of the Membranes

The thermal stability of developed membranes was evaluated by TG analysis. TGA findings for pure PSU and nanocomposite membranes incorporating functionalized-MSNs are shown in Figure 9a. The small mass loss between 100–400 °C for all the membranes is because of the loss of the adsorbed water or the residual DMF solvent within the membranes. The polymer combustion occurs at temperatures of 400–600 °C, which results in the greatest mass loss. The TG curves represent that when the functionalized-MSNs are incorporated, the membrane’s degradation temperature increases, as demonstrated by the change in the TG curves. The incorporation of nanoparticles enhanced the membranes’ thermal stability, as predicted for inorganic-polymer nanocomposite membranes [54,55]. The nanoparticles increase the mass transport barriers’ effect for both oxidizing environment and the volatile compounds generated throughout degradation.

### 3.4. Mechanical Properties of Membrane

The incorporation of functionalized-MSNs into the PSU polymer matrix indicated a substantial trend toward improved mechanical stability as shown in Figure 9b. The findings in several of the cases are extremely encouraging. The pure PSU membrane showed a breaking stress and strain of 6.8 MPa and 0.32, respectively. The functionalized-MSNs/PSU nanocomposite membrane exhibited an enhanced breaking stress and strain of 7.0 MPa and 0.35, respectively. The enhancement in the mechanical properties of the nanocomposite membrane compared to the pure PSU is ascribed to a combination of the PSU matrix to functionalized-MSNs interaction forces.

### 3.5. Equilibrium Water Content and Porosity of Membrane

The equilibrium water content of membranes is an indirect indicator of their hydrophilicity and flux behavior. Figure 10 shows the influence of functionalized-MSNs content on the EWC of functionalized-MSNs/PSU blended membranes. The EWC increased from 61% to 68% when nanoparticles were added to the blending solution, according to the findings. This tendency may be seen in SEM images of the top surface of membranes with nanoparticle compositions. This pattern suggests that functionalized-MSNs interact as an additive, causing nanopores to develop. The EWC is the percentage of water molecules that are available in the nanopores of the nanoparticles. The membrane has become highly porous as the water content has increased. 

Moreover, Figure 10 depicts the porosity of the prepared membrane. Several parameters influence the porosity of a developed membrane, including the polymer material, nanoparticle contents, solvent–polymer interactions, initial drying duration, and water bath temperature, among others. Membrane porosity caused by functionalized-MSNs has a significant impact on membrane surface pore development. The pure PSU membrane has a porosity of 65%. The incorporation of 1.0 wt% nanoparticles enhanced the porosity as well as a mean pore radius of the membrane from 65% to 76% and from 1.3 to 1.9 nm due to the presence of amine and sulphonic functional groups in the solution, that further speed up the transfer rate of solvent and water during in the phase inversion process, allowing the solvent to effectively leach out, resulting in an increased porosity of nanocomposite membrane [56,57].

### 3.6. Membrane Performance

As a function of transmembrane pressure, the pure water permeability of the developed membrane samples was measured. The water permeability of a membrane significantly increases while functionalized-MSNs are incorporated into the PSU polymer, as can be seen in Figure 11. The pure PSU membrane has a permeability of 3.5 LMH bar^−1^, while the functionalized-MSNs/PSU nanocomposite membrane has a permeability of 4.8 LMH bar^−1^. As a result, the enhancement in permeability with regard to functionalized-MSNs contents may be attributed mostly to increased membrane surface hydrophilicity. The existence of the amine and sulphonic groups on the membrane surface, which absorbed water molecules, allowed a rapid flow into the void of the membranes, according to theoretical and experimental study. Furthermore, the improved water permeability of the functionalized-MSNs/PSU nanocomposite, which is influenced by a preferable membrane cross-section structure and produced by continuing to increase the exchange rate between the solvent and non-solvent during in the phase inversion process, decrease membrane mass transfer resistance and formation sponge-like nanostructures [58,59]. The functionalized-MSNs/PSU nanocomposite membrane was able to sustain a wider range of transmembrane pressures with an acceptable permeability, which is ascribed to MSNs good mechanical characteristics.

Three distinct concentrations of heavy metal ions were employed in the simulated feed solutions, which were produced using specified quantities of Cd(NO_3_)^2^·4H_2_O and Zn(NO_3_)^2^ to deionized water, to investigate the influence of heavy metal ion concentrations on the nanocomposite membrane performance. The tests were conducted out at ambient temperature with a consistent transmembrane pressure of 4 bar and at the natural pH of the metal ion solutions. Figure 12 shows the rejection of metal ions and solute fluxes as a function of concentration. The rejections of Cd^2+^ and Zn^2+^ have been somewhat reduced when the feed concentration of an ions is increased while the solute flux is nearly constant for the functionalized-MSNs/PSU nanocomposite membrane. As a result, the optimum heavy metal ion rejection occurs at 5 ppm, with Zn^2+^ rejection of 93% and Cd^2+^ rejection of 90%, respectively. In comparison to concentration, the solute flux remained constant at around 13 ± 2 LMH. There are two possible explanations for these findings. First, because functionalized-MSNs have been included into the nanocomposite membrane, the amine and sulphonic functional groups are readily available on the membrane surface and are responsible for the surface charge. The Donnan exclusion, which is determined by the surface charges of nanocomposite membranes, plays an important role in rejection. Furthermore, in the presence of negative-charged ions such as nitrates, Cd^2+^ and Zn^2+^ maintain the membrane’s electroneutrality, and their hydrated radii greatly increase rejection efficacy, indicating the size exclusion mechanism [10,60,61,62]. During ion rejection, the surface charge and size exclusion both have an impact on the nanocomposite membrane. Second, the ionic strength of the solutions rises as the concentration of the ions increases, and the charge of the membrane surface is impacted. As a result, the electrostatic contact between the membrane and the ionic solution is reduced, and the ions are able to diffuse through the pores of the membrane. On the other hand, the inclusion of a functional group on the nanocomposite membrane surface inhibits concentration polarization while still maintaining hydrophilicity, resulting in a consistent membrane solute flux.

Ion mixture solutions comprising aqueous solution containing 5 ppm quantities of Cd(NO_3_)^2^·4H_2_O and Zn(NO_3_)^2^ salts in deionized water were used to assess the membranes’ performance. The results are shown in Figure 13, and they are evaluated by comparing to the rejection of Cd^2+^ and Zn^2+^ in selective ion solutions. The rejection efficiency of heavy metal ions followed the same sequence as selective ions, with Zn^2+^ rejection of 86% and Cd^2+^ rejection of 80%. However, ion rejection in mixed-ions solutions was lower than in selective ion solutions. Shukla et al. detailed the influence of concentration on ion rejection and the mechanisms that explain it [32,63]. Table 1 summarizes a comparison of functionalized-MSNs/PSU membranes prepared in this study to compare the heavy metal efficiency of nanocomposite membranes in terms of water flux and rejection.

The mixed solution had a greater concentration of divalent cations, which interacted with the membrane surface, lowering its surface charge and enabling ion transport through the membrane pores, which is linked with electrostatic interactions and reduces rejection. Furthermore, the inclusion of anions in the mixed solution facilitated rejection by allowing divalent anions to be excluded more effectively, while maintaining the membrane’s electroneutrality. The finding that the nanocomposite membrane’s surface charge and porosity were more essential for Cd^2+^ and Zn^2+^ rejection is consistent with the Donnan and size exclusion mechanisms, and is confirmed by the experimental results acquired in our investigation. Furthermore, our approach could well be applied to different nanocomposite membranes, allowing for a deeper understanding of heavy metal rejection and the mechanisms behind functionalized-MSNs/PSU nanocomposite membrane separation applications.

## 4. Conclusions

Surface modified with amine and sulphonic groups of mesoporous silica nanoparticles based PSU membranes were fabricated by the phase inversion method for heavy metal ion removal from aqueous solutions containing a wide range of ion concentrations. The resultant nanoparticles and membranes were characterized by their physico-chemical characteristics as well as determination of pure water permeability and cadmium and zinc ion removal. The mesoporous silica nanoparticles’ surface modification had a significant effect on membrane performance. The functionalized mesoporous silica nanoparticles increased the pore size in the sub-layer and enhanced the interconnectivity of pores between the sub-layer and the bottom layer, according to the SEM data. The incorporation of these nanoparticles enhanced the membrane porosity, water content, thermal stability, and mechanical strength. According to the filtration performance findings, the nanocomposite membrane exhibited greater pure water permeability than a pure PSU membrane. The pure water permeability of the nanocomposite membrane with 1.0wt% of nanoparticles reached 4.8 LMH bar^−1^. In addition, the inclusion of amine and sulphonic functional groups on mesoporous silica nanoparticles increased the effectiveness of developed membranes to remove cadmium and zinc by more than 90%. As a result, this research presents a membrane separation method that is impressively practical, economically viable, and efficient, with prospective applications in wastewater treatment and long-term separation.

## Figures and Tables

**Figure 1 membranes-11-00935-f001:**
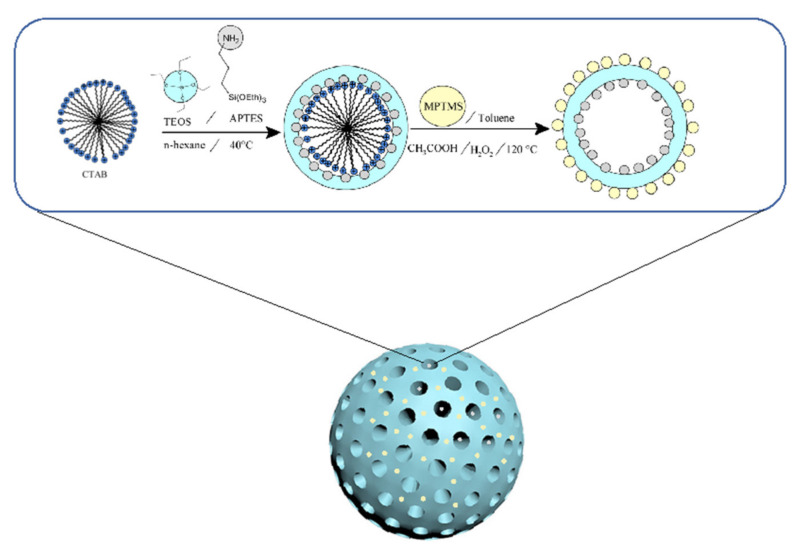
Schematic diagram of the synthesis procedure for the functionalized mesoporous silica nanomaterial.

**Figure 2 membranes-11-00935-f002:**
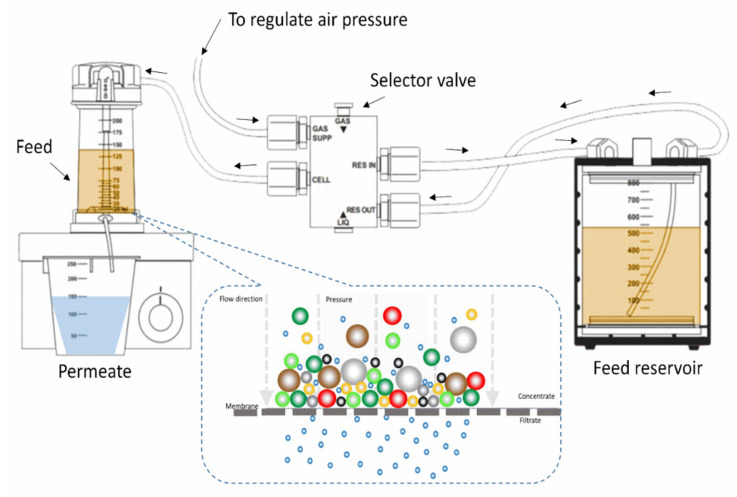
Schematic diagram of the dead-end stirred-cell filtration setup.

**Figure 3 membranes-11-00935-f003:**
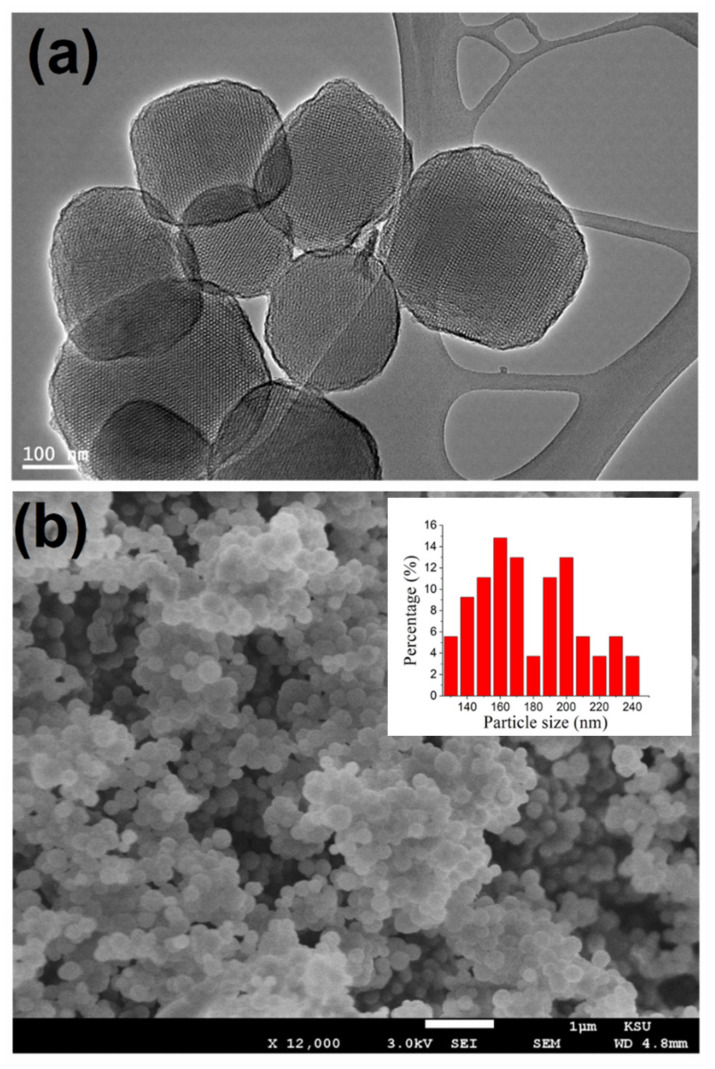
(**a**) TEM and (**b**) SEM images of functionalized mesoporous silica nanoparticles, the inset shows the statistical diameter distribution profile of 125 NP observed in the SEM study using ImageJ software.

**Figure 4 membranes-11-00935-f004:**
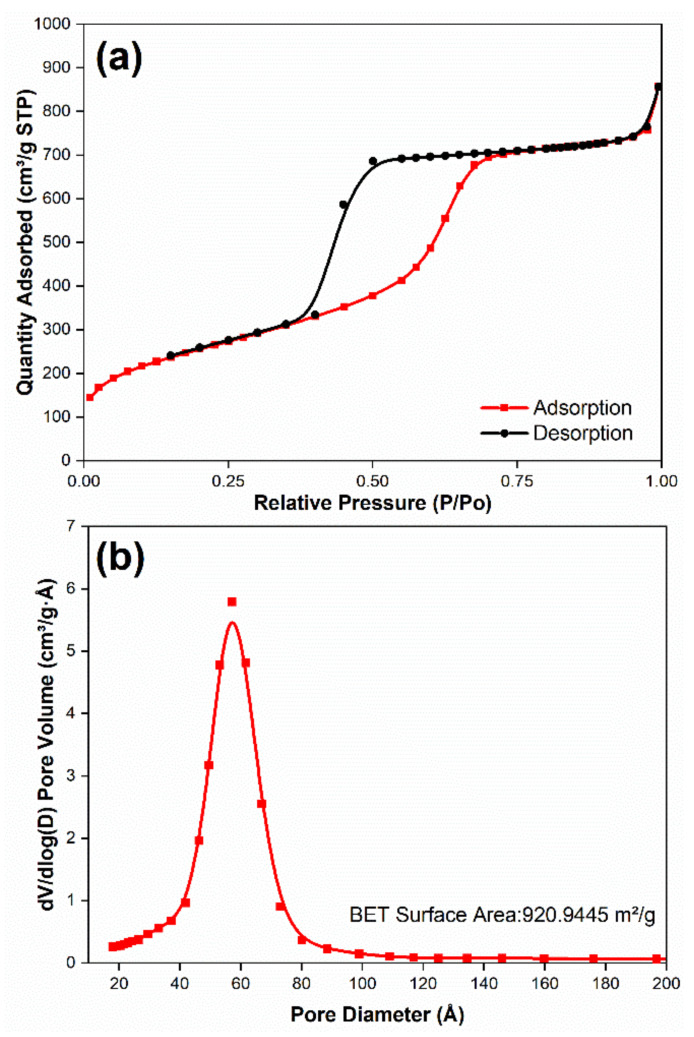
Functionalized mesoporous silica nanocomposites: (**a**) BET nitrogen adsorption–desorption isotherms and (**b**) the pore size distribution plots.

**Figure 5 membranes-11-00935-f005:**
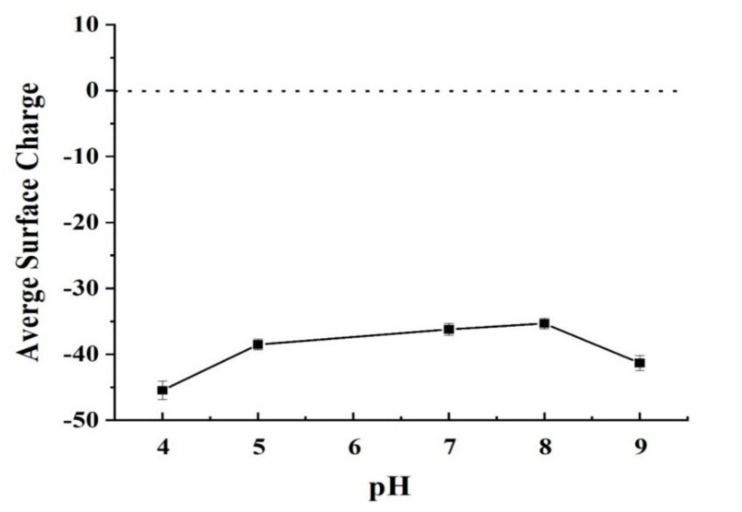
Surface charging properties of the functionalized mesoporous silica nanoparticle.

**Figure 6 membranes-11-00935-f006:**
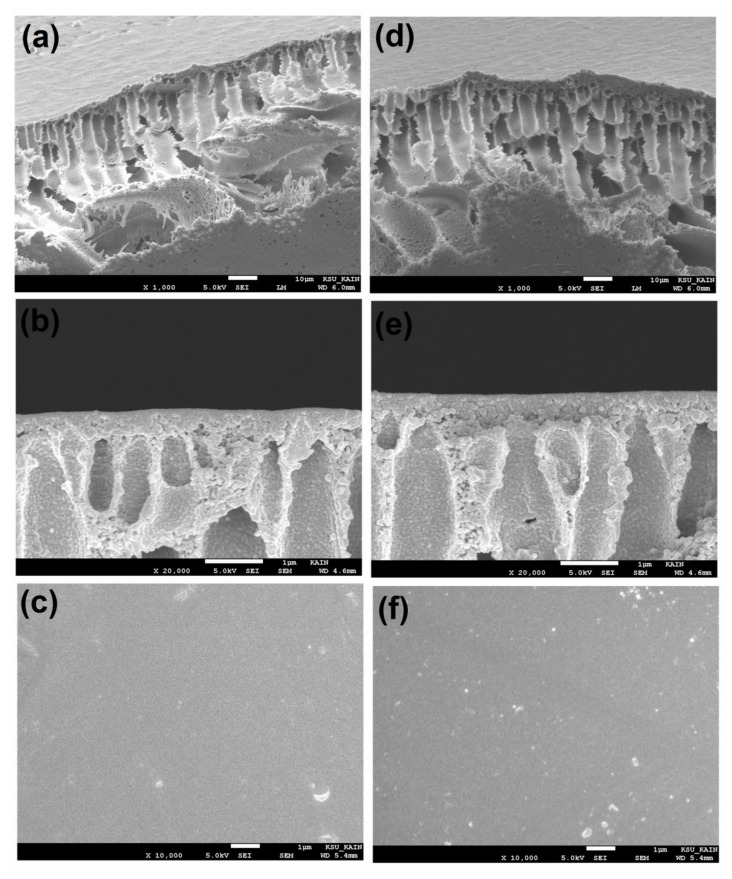
Cross-sectional SEM images of the (**a**,**b**) pure PSU, (**d**,**e**) functionalized-MSNs/PSU nanocomposites membranes, and (**c**,**f**) surface SEM images of pure PSU and functionalized-MSNs/PSU nanocomposites membranes.

**Figure 7 membranes-11-00935-f007:**
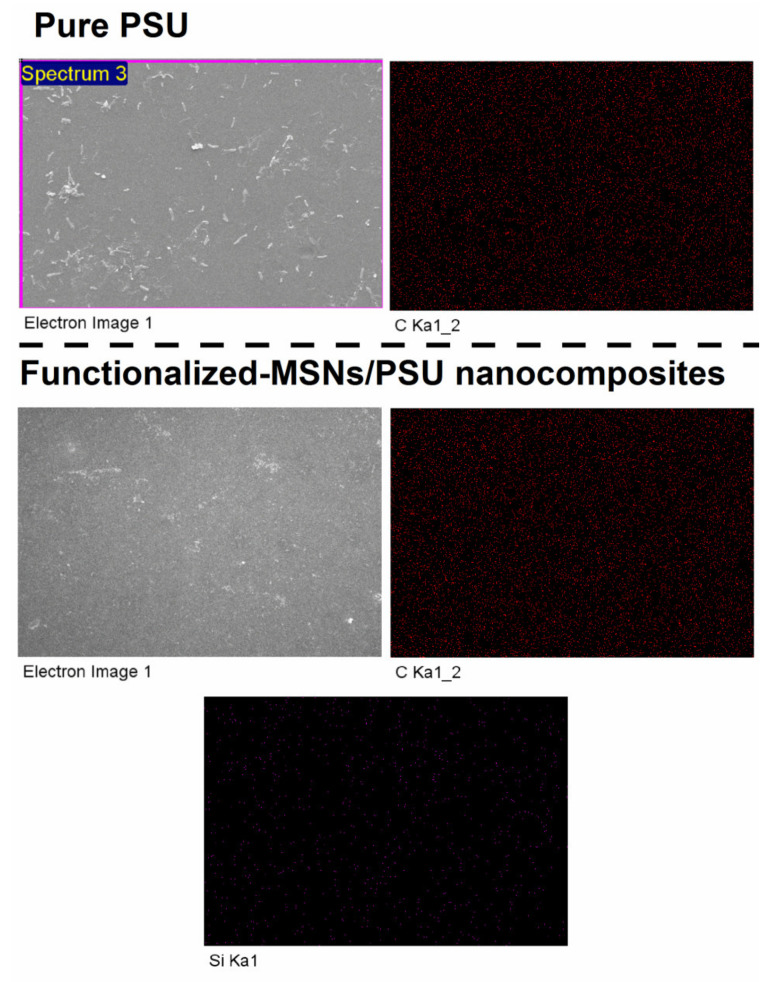
Elemental mapping on the membrane surface.

**Figure 8 membranes-11-00935-f008:**
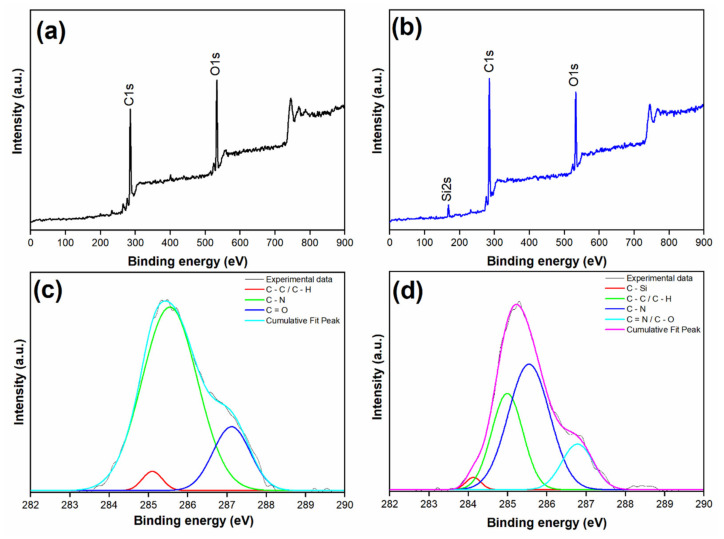
XPS spectra of the membranes for the binding energies: (**a**) PSU and (**b**) functionalized-MSNs/PSU, and XPS high-resolution spectra of C1s in (**c**) PSU and (**d**) functionalized-MSNs/PSU.

**Figure 9 membranes-11-00935-f009:**
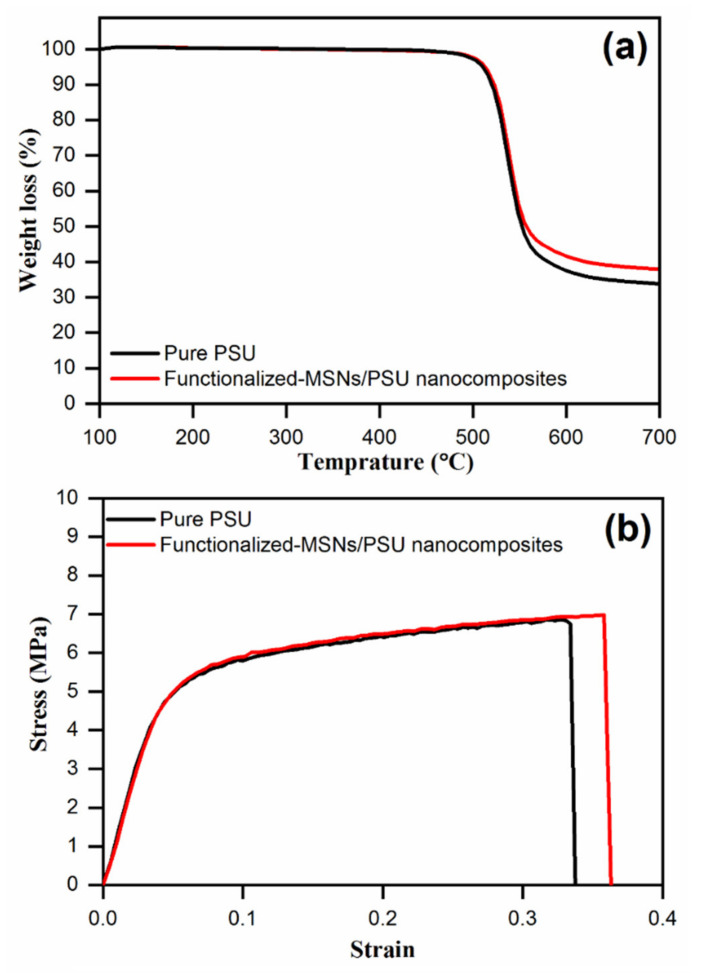
(**a**) TGA and (**b**) tensile stress–strain curves of the pure PSU and functionalized-MSNs/PSU nanocomposite membranes.

**Figure 10 membranes-11-00935-f010:**
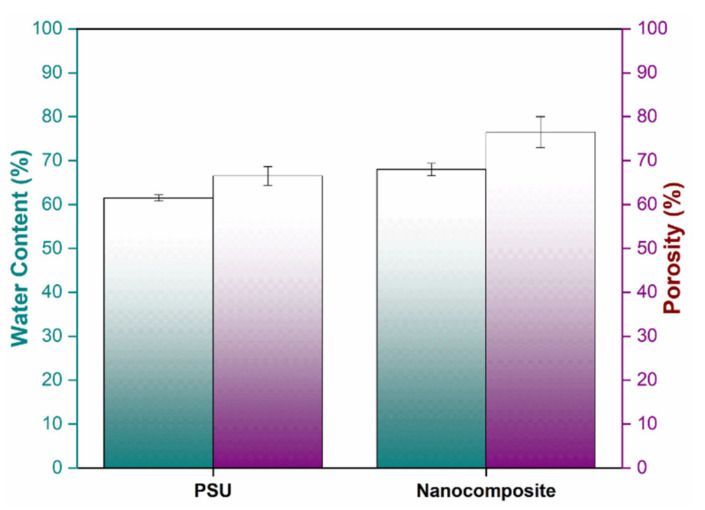
Pure PSU and functionalized-MSNs/PSU nanocomposite membrane water content and porosity.

**Figure 11 membranes-11-00935-f011:**
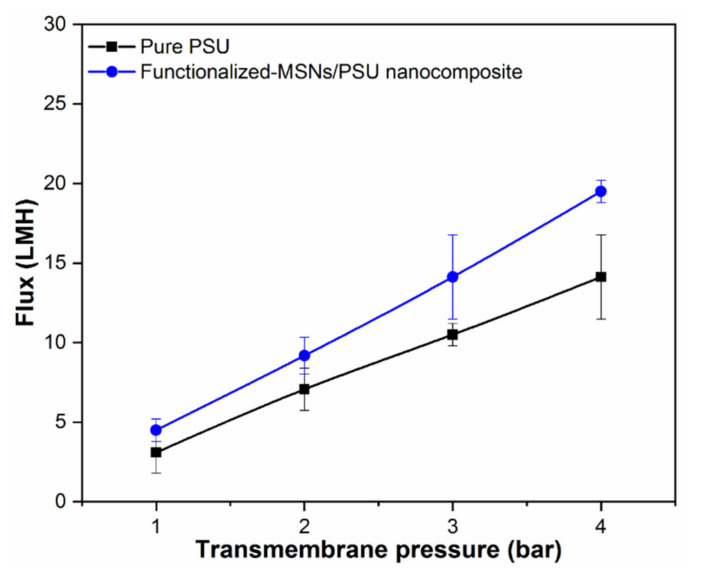
Pure PSU and functionalized-MSNs/PSU nanocomposite membrane performance in terms of pure water permeabilities (water flux versus applied TMP at 1 to 4 bar).

**Figure 12 membranes-11-00935-f012:**
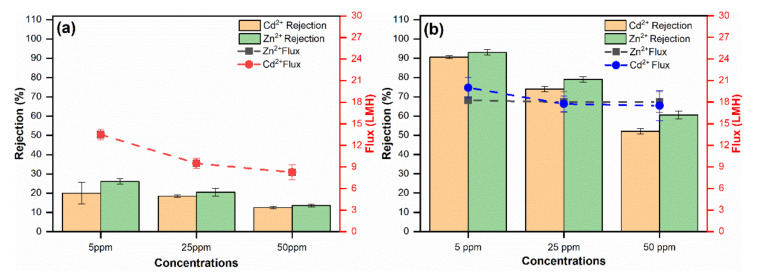
Removal performance of (**a**) pure PSU membrane rejection and flux, (**b**) functionalized-MSNs/PSU nanocomposite membrane rejection and flux of membranes toward Cd(NO_3_)^2^·4H_2_O, and Zn(NO_3_)^2^ as a function of ionic concentration (5, 25, and 50 ppm) at 4 bar, room temperature with natural pH.

**Figure 13 membranes-11-00935-f013:**
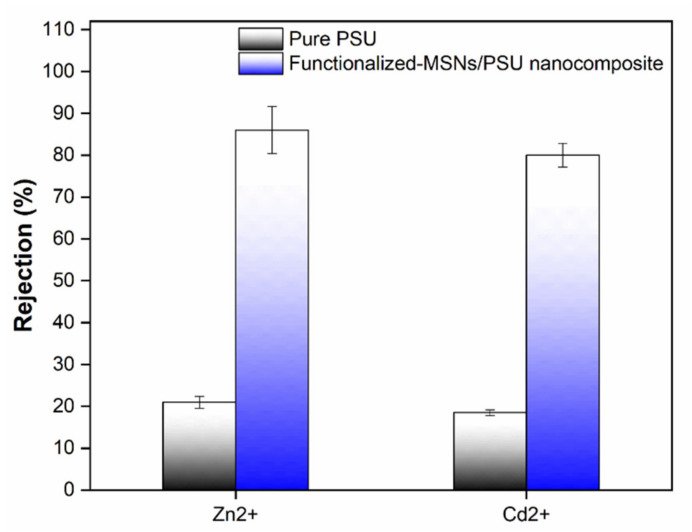
Ion rejection in the mixture of Cd(NO3)_2_·4H_2_O, and Zn(NO_3_)_2_ solution each with concentrations of 5 ppm under 4 bar pressure at natural pH.

**Table 1 membranes-11-00935-t001:** Performance comparisons of membranes prepared in this study versus membranes reported previously.

Membranes	Water Flux LMH	Rejection (%)		References
		Cd^2+^	Zn^2+^	
PPSU/MWCNTs	185	72	-	[64]
PDA/MOF-TFN	11	82	-	[65]
Matrimid/PEI	2.9	97	98	[66]
PSU/sPPSU	190	69	-	[67]
PAN/MOF-808	348	57	64	[68]
eGO3/PA-HFC	18	-	93	[69]
Polyamide NF	40	84	87	[70]
Functionalized-MSNs/PSU	20	91	94	This work

## Data Availability

The data presented in this study are available on request from the corresponding author.

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
