# Peer review of "Fabrication of Polysulfone-Surface Functionalized Mesoporous Silica Nanocomposite Membranes for Removal of Heavy Metal Ions from Wastewater"

_membranes, 2021, doi:10.3390/membranes11120935_

Round 1

Reviewer 1 Report

This work focused on preparation of functionalized-MSNs/PSU nanocomposite membranes for heavy metal removals. The modified membrane exhibits better performance than that of pristine PSf membrane. However, the manuscript needs major revision before publishing in Membranes. The following are my concerns:

  1. In the introduction, please mention what type of separation process used in this work if ultrafiltration or nanofiltration membrane. Also, please discuss its advantages over traditional method.
  2. Schematic diagram/reaction mechanism of functionalization of MSN, and also membrane fabrication is suggested to include in the manuscript.
  3. Figure 4b-e, the membrane looks deformed. It is suggested to use better SEM images.
  4. Please provide elemental mapping on the membrane surface to prove the dispersion of the MSN on the membrane. EDX analysis can provide this mapping.
  5. How the change in concentration of functionalize MSN affects the membrane property and performance. Please give a proof.
  6. From the porosity data, please calculate the mean pore radius of the membranes.
  7. How was the functionalized MSN affects the membrane MWCO?
  8. Please provide the information of membrane surface zeta potential. Since the authors claimed that “The finding 408 that the nanocomposite membrane's surface charge and porosity were more essential for Cd2+ and Zn2+ rejection is consistent with the Donnan and size exclusion mechanisms, and is confirmed by the experimental results acquired in our investigation.”
  9. Please provide a comparison study of this work from other literature.
  10. How was the stability and antifouling property of the membrane?

Author Response

Point-by-Point Response to Reviewers

Reviewer #1

Query (1): In the introduction, please mention what type of separation process used in this work if ultrafiltration or nanofiltration membrane. Also, please discuss its advantages over traditional method.

Response: We thank the reviewer for the accurate summary of our work, and especially for positive comments on the originality and quality of our research work and appreciation of the importance of the findings. As per the reviewer's suggestion, the manuscript is revised with regard to the particular aspect specified. In the revised manuscript, the separation process and its advantages over traditional method in this work are included.

Query (2): Schematic diagram/reaction mechanism of functionalization of MSN, and also membrane fabrication is suggested to include in the manuscript.

Response: Thank you for taking care of our manuscript deeply. The Schematic diagram/reaction mechanism of functionalization of MSN, and membrane fabrication has been included in the revised manuscript.

Query (3): Figure 4b-e, the membrane looks deformed. It is suggested to use better SEM images.

Response: The reviewer has correctly pointed out about Figures 4b-e (now 6b-e). In the revised manuscript, we have added the better SEM images.

Query (4): Please provide elemental mapping on the membrane surface to prove the dispersion of the MSN on the membrane. EDX analysis can provide this mapping.

Response: Thank you very much for the nice and valuable suggestion, in revised manuscript, we have added to elemental mapping on the membrane surface to prove the dispersion of the MSN on the membrane.

Query (5): How the change in concentration of functionalize MSN affects the membrane property and performance. Please give a proof.

Response: The authors would like to express their gratitude to the reviewer for this insightful comment. We fully agree with the reviewer's assessments that the concentration of functionalize MSN affects the membrane property and performance due to the physicochemical properties of nanoparticles. The specific concentration of nanoparticles allowed for more exposure of the surface and its functional groups to the surrounding area, resulting in improved dispersion in the polymer solution and reduced aggregation. Therefore, the nanocomposite membrane has better performance and results in a manuscript.

Query (6): From the porosity data, please calculate the mean pore radius of the membranes.

Response: Thank you, it has been done in revised manuscript.

Query (7): How was the functionalized MSN affects the membrane MWCO?

Response: Thank you so much for your wonderful feedback. It is well known that incorporating hydrophilic nanoparticles to the casting solution can accelerate the exchange rate between solvent and nonsolvent during the coagulation process, resulting in a porosity structure, however this does not increase pore size and functionalized MSN has no effect on the membrane MWCO.

Query (8): Please provide the information of membrane surface zeta potential. Since the authors claimed that “The finding 408 that the nanocomposite membrane's surface charge and porosity were more essential for Cd2+ and Zn2+ rejection is consistent with the Donnan and size exclusion mechanisms, and is confirmed by the experimental results acquired in our investigation.”

Response: Thank you for your thoughtful note on the membranes' surface zeta potential. In response to this statement, we have added and clarified the surface charge scenario in the revised manuscript.

Query (9): Please provide a comparison study of this work from other literature.

Response: Thank you very much for the nice and valuable suggestion. In the revised manuscript, we have added the comparison table for the performance of functionalized-MSNs/PSU nanocomposites membranes and previously reported studied with this work.

Query (10): How was the stability and antifouling property of the membrane?

Response: Good comment, functionalized-MSNs/PSU nanocomposites membranes have high aqueous stability due to their key structural property. In general, a water stable functionalized-MSNs structure must be strong enough to withstand water molecules attacking the strong bonds formed. Moreover, we conducted different pressure filtrations and separation studies with membranes. We found that the results of filtration and rejection were consistent across all replications of the experiments. Due to the presence of amine and sulphonic groups of the nanoparticles on the membrane surface the membrane have excellent antifouling properties of the nanocomposite membrane. The functional groups increased the surface charge of the membrane and enhanced its hydrophilicity. The surface charge and hydrophilicity of the surface effectively decreased the fouling via the adsorption of water molecules and the subsequent increase in the electrostatic repulsion between the surfaces of membrane and protein, which hindered cake formation and adsorption of foulants onto the membrane surface

We hope that our revised manuscript is now considered acceptable for publication in Membranes journal.

Reviewer 2 Report

The current study test novel MSNs-modified membranes for the removal of heavy metals. This manuscript does a great job in terms of membrane/MSNs characterization (i.e., porosity, water flux, rejection), as well as testing these membranes for the removal of 2 selected heavy metals. In addition, the dominant mechanisms for the rejection of these heavy metals are proposed. Although comprehensive and sound, the manuscript is missing the key information for potential publication in this reputed journal. Specifically, I observe there is a link missing between the results obtained and potential applications/big picture. As a result, the paper is read as a generic material testing report. Below my specific comments.  

  1. Why were Cadmium and Zinc were selected? Why not arsenic, chromium, lead, etc. What is the rationale behind this selection? Is it related to specific industrial activity? (the paper lacks justification)
  2. Would the authors expect similar rejections with arsenic, chromium, lead, etc.? why? why not? (hint: does valence, radii, etc, play a role?)
  3. The size of the nanoparticles was reported as ~165 nm and homogeneous. What is the stdev of this diameter?
  4. Why were these concentrations (5, 25 and 50 ppm) selected? Based on what scenario? Would the membranes behave similarly at heavy metal concentrations at the level of ug/L?
  5. These experiments were conducted under extremely controlled conditions (synthetic streams). Under real conditions, you would expect foulants. What are the potential foulants to these types of membranes?
  6. When addressing the points below, you can build a better discussion that includes an “implications section”, where potential niches of use can be discussed.

Author Response

Reviewer #2

Query (1): Why were Cadmium and Zinc were selected? Why not arsenic, chromium, lead, etc. What is the rationale behind this selection? Is it related to specific industrial activity? (the paper lacks justification).

Response: The authors would like to express their gratitude to the reviewer for taking an interest in this topic. We completely agree with the reviewer's observation about cadmium and zinc selection. As we know that the presence of these heavy metals in wastewaters causes many problems due to their persistence in the environment. There is also a high risk of heavy metal accumulation in the body tissues of living organisms due to their high solubility in the aquatic environments. Heavy metals are usually produced by industries, which are regarded to be the most harmful industrial effluent. These are a very dangerous agent due to its side effects on the human body even at concentrations as low. Other heavy metals, such as arsenic, chromium, lead, and others, which are regularly present in wastewaters over their acceptable level, are also extremely dangerous. For this heavy metal, we will do research in the near future to better understand membrane performance.

Query (2): Would the authors expect similar rejections with arsenic, chromium, lead, etc.? why? why not? (hint: does valence, radii, etc, play a role?).

Response: Thank you for the comment and the author agrees absolutely with the reviewer's comment that the rejections with arsenic, chromium, lead etc. The prepared nanocomposite membrane has nanoporous structure as per morphological study. Thereby authors also expect similar rejections with arsenic, chromium, lead, etc. due to their valence, radii, etc, and represent the size exclusion mechanism.

Query (3): The size of the nanoparticles was reported as ~165 nm and homogeneous. What is the stdev of this diameter?

Response: Thank you for your comment. In the revised manuscript, the stdev of the nanoparticles diameter was added.

Query (4): Why were these concentrations (5, 25 and 50 ppm) selected? Based on what scenario? Would the membranes behave similarly at heavy metal concentrations at the level of ug/L?

Response: The authors would like to express their gratitude to the reviewer for taking an interest in this topic. According to US EPA guidelines, the maximum contaminant level (MCL) in water is 0.005 ppm for cadmium and 5 ppm for zinc. If the nanocomposite membrane rejects a larger concentration of the ions listed above, this is common in industrial wastewater, so we expect that the membranes behave similarly at heavy metal concentrations at the level of ug/L.

Query (5): These experiments were conducted under extremely controlled conditions (synthetic streams). Under real conditions, you would expect foulants. What are the potential foulants to these types of membranes?

Response: Thank you for taking care of our manuscript deeply. We fully agree with the reviewer's assessments, as we know that the fouling is generally initiated by the growth and deposition of the foulants on membrane surface. Foulants include contaminating microorganisms, inorganic or organic compounds, and colloids. The nanocomposite membrane has good antifouling capabilities due to the presence of amine and sulphonic groups of nanoparticles on the membrane surface. The functional groups boosted the membrane's surface charge and improved its hydrophilicity. The surface charge and hydrophilicity substantially reduced fouling via the adsorption of water molecules and the consequent increase in electrostatic repulsion between the membrane and protein surfaces, preventing cake formation and foulant adsorption onto the membrane surface.

Query (6): When addressing the points below, you can build a better discussion that includes an “implications section”, where potential niches of use can be discussed.

Response: Thank you for taking such great care of our manuscript. In the revised manuscript, the implications of our research finding were included.

We hope that our revised manuscript is now considered acceptable for publication in Membranes.

Round 2

Reviewer 1 Report

The revised manuscript has been reviewed again. All suggestions have appropriately corrected. It can be considered for acceptance.